# Radiological Features of Microvascular Invasion of Hepatocellular Carcinoma in Patients with Non-Alcoholic Fatty Liver Disease

**Matteo Renzulli [1,*][ID], Anna Pecorelli [1], Nicolò Brandi [1,*][ID], Giovanni Marasco [2][ID], Francesco Adduci [1], Francesco Tovoli [3][ID], Bernardo Stefanini [3], Alessandro Granito [3][ID] and Rita Golfieri [1][ID]**

1   Department of Radiology, IRCCS Azienda Ospedaliero-Universitaria di Bologna, Via Albertoni 15, 40138 Bologna, Italy
2   Internal Medicine and Digestive Physiopathology Unit, IRCCS Azienda Ospedaliero-Universitaria di Bologna, 40138 Bologna, Italy
3   Division of Internal Medicine, IRCCS Azienda Ospedaliero-Universitaria di Bologna, 40138 Bologna, Italy
*   Correspondence: matteo.renzulli@aosp.bo.it (M.R.); nicolo.brandi@studio.unibo.it (N.B.)

**Abstract: Background:** The aim of the present study was to evaluate the presence and the prognostic value of the radiological signs of microvascular invasion (MVI) of hepatocellular carcinoma (HCC) in patients with non-alcoholic fatty liver disease (NAFLD). **Methods:** Between January 2015 and December 2017, all patients (91 patients) with de novo HCC or HCC recurrence occurring at least 2 years after the last treatment in NAFLD (36 patients) or with hepatitis C virus (HCV) liver disease (55 patients) were included. Each HCC was treated with liver resection and transplantation to obtain the anatomopathological confirmation of MVI. All patients had at least one available computed tomography (CT) scan or magnetic resonance imaging (MRI) performed no more than one month prior to the treatment. The clinical data of each patient, tumor burden (diameter, margins, two-trait predictor of venous invasion (TTPVI), and peritumoral enhancement), the recurrence rate (RR) after a 1-year follow-up, and the time to recurrence (TTR) were collected. **Results:** The NAFLD–HCC nodules were larger as compared to HCV–HCC (51 mm vs. 36 mm, $p = 0.004$) and showed a higher prevalence of TTPVI (38.9 vs. 20.0%, $p = 0.058$). At multivariate analysis, nodule diameter >50 mm was found to be the only independent prognostic factor of TTPVI (hazard ratio: 21.3, 95% confidence interval: 4.2–107.7, $p < 0.001$), and the presence of TTPVI was confirmed to be the only independent prognostic factors of recurrence (hazard ratio: 2.349, 95% confidence interval: 1.369–4.032, $p = 0.002$). No correlations were found between TTR and irregular tumor margins or peritumoral enhancement. **Conclusion:** The NAFLD–HCC patients had larger tumors at diagnosis and showed a more frequent presence of radiological signs of MVI as compared to the HCV–HCC patients. The MVI was related to a more rapid recurrence after curative treatments, demonstrating the prognostic value of this radiological diagnosis.

**Keywords:** non-alcoholic fatty liver disease; hepatitis C; hepatocellular carcinoma; microvascular invasion; computed tomography; magnetic resonance imaging

## 1. Introduction

Hepatocellular carcinoma (HCC) is the most frequent primary malignant tumor of the liver [1]. It represents the fourth leading cause of cancer-related death worldwide, accounting for approximately 800,000 deaths/year [2]. An elevated number of risk factors for developing HCC has been identified, including hepatitis B virus (HBV) infection, chronic hepatitis C virus (HCV) infection, and cirrhosis from all causes [3]. In Western countries, the most commonly observed risk factor for HCC is HCV infection, followed by alcohol use and non-alcoholic fatty liver disease (NAFLD) [4]. The prevalence of NAFLD, which represents the hepatic manifestation of the metabolic syndrome, is rapidly growing,

especially in Western countries, in parallel with widespread obesity and type 2 diabetes mellitus [5]. It includes a wide spectrum of disorders, ranging from hepatic steatosis (NAFL) to non-alcoholic steatohepatitis (NASH) [6], which can only be differentiated by liver biopsy. Contrary to NAFL, in fact, NASH is characterized by hepatic steatosis associated with liver [7]. The diagnosis of NAFLD requires the demonstration of a fatty liver by imaging or biopsy after excluding significant alcohol consumption, other possible causes of fatty liver disease, and the absence of coexisting chronic liver disease [8,9]. Patients with NAFLD are at high risk for liver fibrosis, cirrhosis, and HCC [9–11]; NAFLD probably accounts for a significant portion of the cirrhosis classified as cryptogenic [12]. The most frequent type of cancer associated with type 2 diabetes has been shown to be HCC [13], and obesity almost doubles the risk of HCC [14,15]. Therefore, a rapidly increasing incidence of NAFLD–HCC might be expected due to the increase in type 2 diabetes mellitus and obesity in Western countries [16]. Moreover, NAFLD–HCC can also arise in the absence of cirrhosis [5].

In cirrhotic or high-risk patients, HCC can be diagnosed by imaging alone using dedicated protocols, obviating the need for biopsy [17–19]. This possibility, unique in the field of oncology for solid tumors, is due to the peculiarity of the hepatic vascularization and the tumor vascular characteristics [20]. One of the major problems regarding HCC curative treatments is the unsatisfactory overall survival rate, related to the high recurrence rate, regardless of the etiology of the liver disease. A five-year recurrence of HCC occurs in 25% of cases after liver transplantation and in 70% of cases after hepatic resection [21]. In this regard, one of the most relevant prognostic factors is vascular invasion [22]. Macrovascular invasion and microvascular invasion (MVI) of HCC are related to a 15-fold and a 4.4-fold increased risk of tumor recurrence, respectively [23]. Microvascular invasion is defined as a microscopic tumor invasion of the small intrahepatic vessels, including those of the portal vein, hepatic artery, and ductal system [24]. Moreover, MVI is a well-known sign of a poor prognosis, which, unlike macrovascular invasion, is extremely difficult to predict directly using conventional imaging methods such as computed tomography (CT) and magnetic resonance imaging (MRI) [25]. In addition, the detection of MVI using preoperative biopsy has proven to be unreliable due to sampling error caused by tumor heterogeneity [26]. Thus, MVI can only be diagnosed after surgical treatment by histopathological evaluation; therefore, due to this late postoperative diagnosis, it has limited usefulness in current clinical practice, such as in making therapeutic decisions [27]. It is worth emphasizing that MVI is one of the most important predictors of early recurrence, so-called true recurrence, which occurs within two years after the curative treatment [28]. In order to obtain a preoperative prediction of MVI, much effort has been made in the radiological field to identify the imaging features predictive of MVI in HCC. In particular, some "worrisome" imaging features, such as (1) nonsmooth tumor margins, (2) peritumoral enhancement, (3) two-trait predictor of venous invasion (TTPVI; "internal arteries" and "hypoattenuating halos"), and (4) large tumor size, have been identified as significant predictors of the presence of MVI in HCC [29]. In the setting of NAFLD, the investigation of MVI by imaging is even more interesting. In fact, it would be helpful to understand whether NAFLD–HCC is biologically more aggressive than other forms of HCC or whether the worst prognosis depends on the diagnosis at a more advanced stage due to the absence of a stringent surveillance program.

The aim of the present study was to evaluate the presence and the prognostic value of the radiological signs of MVI of HCC in patients with NAFLD.

## 2. Materials and Methods

This was a single-center retrospective observational study approved by the Institutional Review Board. Written informed consent was obtained from all the patients, and the study was conducted in compliance with the Declaration of Helsinki for clinical studies.

The inclusion criteria were: (1) age >18 years old; (2) de novo HCC or HCC recurrence occurring at least two years after the last treatment in both NAFLD (cases) and HCV liver disease (controls), diagnosed according to the European Association of the

Study of the Liver (EASL) guidelines in the period between January 2015 and December 2017 [6]; (3) treatment with curative therapies (liver resection and transplantation); (4) tanatomopathological confirmation of MVI; and (5) available computed tomography (CT) scan or magnetic resonance imaging (MRI) performed no more than one month prior to the treatment.

Patients with unavailable histopathological confirmation of MVI (*n* = 8), de novo HCC or HCC recurrence occurring during the two years after the last treatment (*n* = 40), or an inadequate imaging study (*n* = 10) were excluded from the study (Figure 1).

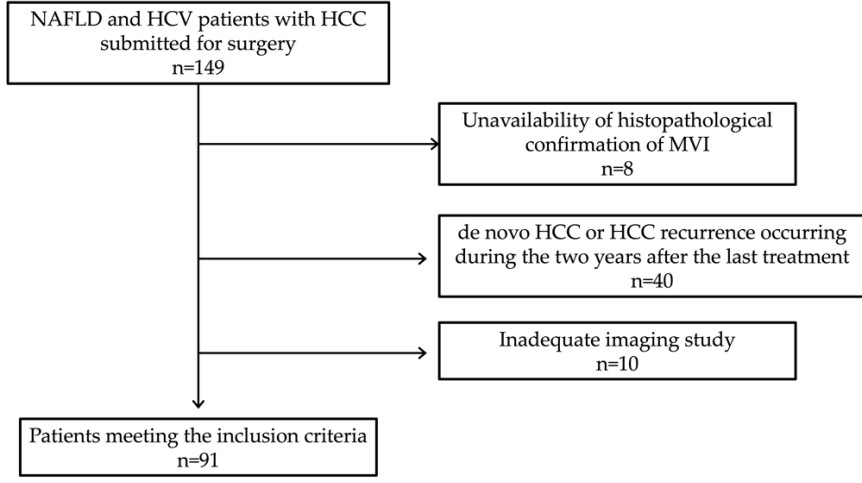

**Figure 1.** Flow diagram of patient selection in the study.

Based on the above criteria, the final study population consisted of 36 cases and 55 controls. The following variables were collected for each patient: age at diagnosis, gender, body mass index (BMI), neoplastic characteristics (number of nodules and diameter of the largest nodule), neoplastic stage classified according to the Barcelona Clinic Liver Cancer (BCLC) classification, recurrence rate (RR) after a follow-up of 12 months, and time to recurrence (TTR) defined as the time from the curative treatment to disease recurrence.

All the pathologic examinations were performed by a team of pathologists, each one with more than 15 years of experience in liver pathology. MVI was defined as a tumor within a vascular space lined by the endothelium, which was visible only at microscopy [30]. The MVI of tumor cells into the portal or hepatic venules and capillaries was pathologically examined by sampling the HCC tissue.

*2.1. Imaging Analysis*

All the CT or MRI images were acquired according to standard protocols recommended by the international guidelines and previously described [19,29,31]. In particular, CT was performed by using a 64-section multidetector CT scanner (Lightspeed VCT 64; GE Healthcare, Milwaukee, WI, USA). The procedure was performed at baseline and after the administration of 150–180 mL of the tri-iodinated nonionic contrast agent iomeprol (Iomeron, Bracco, Milan, Italy, (350 mg iodine per mL)) at a flow rate of 3–4 mL/s into an antecubital vein by using an automated power injector. The examination was performed by using a multiphasic technique and bolus tracking at three different phases: late arterial phase (after 25–30 s), venous phase (after 45–60 s), and delayed phase (after 180–300 s). MR imaging was performed by using a 1.5 T superconducting system (Signa; GE Medical Systems, Milwaukee, WI, USA) with a body phased-array multicoil for signal detection. Unenhanced sequences were as follows: (a) a respiratory-triggered fat-suppressed fast recovery T2-weighted fast spin-echo sequence (repetition time (ms)/echo time (ms), 1500–1760/100; 256 × 190 matrix; 41.67 kHz per pixel bandwidth; and 20–25 s acquisition time) or a breath-hold T2-weighted single-shot fast spin-echo sequence (2.000/80, matrix of 256/190), with and without fat saturation; and (b) breath-hold T1-weighted gradient-echo

dual echo "in and out of phase" sequence (150/4.6 and 150/2.1, respectively; 80 flip angle; $256 \times 160$ matrix; 62.50 Hz per pixel bandwidth; one signal acquired; and 20–25 s acquisition time). The contrast-enhanced pulse sequences were performed after the intravenous injection of gadolinium ethoxybenzyl diethylenetriamine pentaacetic acid (EOB Primovist; Bayer Schering Pharma, Berlin, Germany) at a dose of 0.1 mL per kg of body weight at a speed of 1 mL/s, immediately followed by a 20 mL saline flush through an antecubital venous catheter by using a dual power injector. The images were obtained by using a fat-suppressed three-dimensional gradient-echo sequence (liver acquisition with volume acceleration) before and after gadolinium ethoxybenzyl diethylenetriamine pentaacetic acid administration. The hepatic arterial phase images were obtained 7 s after the arrival of the contrast medium in the distal thoracic aorta, and the venous and equilibrium phase images were obtained after 60 and 180 s, respectively. All the images were evaluated by a single radiologist with 15 years of experience in liver imaging who was blind to inclusion criteria, in particular to the presence of MVI. The following criteria for identifying the radiological signs of MVI were evaluated: (1) nodular diameter (expressed as mean and standard deviation); (2) nodular margins (categorized as smooth or irregular); (3) TTPVI (categorized as present or absent); and (4) peritumoral enhancement (categorized as present or absent).

The nodular margins were defined as irregular when the tumor was non-nodular in all imaging planes.

The TTPVI was the result of a radiogenomic algorithm based on the association between two imaging patterns and the expression variation of 91 genes constituting the "genetic signature of microscopic venous invasion" in many types of cancer, in particular in HCC. The two imaging patterns of MVI associated with genetic alterations are the presence of internal arteries inside the tumor and the absence or non-continuity of a hypodense halo. Peritumoral enhancement was represented by a detectable hyperenhancing portion in the arterial phase, adjacent to the tumor border, which becomes isodense in CT images and isointense in MR images with respect to the surrounding liver parenchyma [29].

### 2.2. Statistical Analysis

The distribution of continuous variables was assessed using the Kolmogorv–Smirnov test. Categorical variables were expressed as numbers and percentages and were compared using the Fisher exact test. Continuous variables were expressed as mean and standard deviation and were compared using Student's *t* test, with the exception of age, which was expressed as median and interquartile range (IQR) for descriptive purposes. Correlations with *p* values < 0.05 were considered to be statistically significant. Times to recurrence were analyzed using the Kaplan–Meyer method, and the curves were compared using the log-rank test. Univariate analysis was carried out to assess the degree of association between recurrence and the radiological signs of MVI. Variables associated with recurrence ($p \leq 0.10$) were tested using the Cox multivariate regression model. The hazard ratio (HR) and 95% confidence interval were calculated for the independent predictors of recurrence. All the analyses were carried out using the SPSS 21.0 statistical package (SPSS Incorporated Chicago, Chicago, IL, USA).

### 3. Results

Most NAFLD–HCC patients were male (*n*= 31; 86.8%), with a median age of 69 years (IQR 50–86). In the majority of these patients, the HCC developed as a monofocal lesion (*n* = 22; 61.2%) at the early stage (*n* = 22; 61.2%). In HCV–HCC patients, the median age was almost the same (median 70 years; IQR 40–87); however, the prevalence of males was slightly less (74.1% vs. 86.8%, *p* = 0.749). The BMI was 28.6 kg/m$^2$ ($\pm$4) in NAFLD–HCC patients and 27.1 kg/m$^2$ ($\pm$4.1) in HCV–HCC patients, with no significant difference between the groups (*p* = 0.871).

All of the patients included in the present study (*n* = 91) received anatomopathological confirmation of MVI. In particular, 25 patients (70%) of the NAFLD–HCC group underwent

resection, whereas 11 patients (30%) underwent liver transplantation (LT); in the HCV–HCC group, 40 patients (73%) underwent resection and 15 (27.2%) LT (*p* = 1.000). Treatment modality was considered in the uni- and multivariate analyses; however, it was not selected as an independent prognostic factor of recurrence in our population.

At the time of diagnosis, the HCC appeared more frequently as multinodular disease (*n* = 32; 58% vs. 14; 38.8%) at the early stage (*n* = 33; 60%) (Table 1; Figures 2 and 3).

**Table 1.** Characteristics of the study population.

| Variable | NAFLD (*n* = 36) | HCV (*n* = 55) | *p* |
|---|---|---|---|
| Males | 31 (86.8%) | 41 (74.1%) | 0.749 |
| Age (years) | 69 (50–86) | 70 (40–87) | 1.000 |
| BMI (kg/m$^2$): mean ± SD | 28.6 (±5) | 27.1 (±4.1) | 0.871 |
| Diameter (mm): mean ± SD | 51 (±23) | 26 (±16) | <0.0001 |
| Multinodularity | 14 (38.8%) | 32 (58.0%) | 0.349 |
| BCLC | | | |
| Very early (0) | 14 (38.8%) | 22 (40%) | 0.882 |
| Early (A) | 22 (61.2%) | 33 (60%) | |
| Child–Pugh | | | |
| Class A5 | 15 (41.7%) | 24 (43.6%) | 0.798 |
| Class A6 | 21 (58.3%) | 31 (56.4%) | |
| Milan in | 30 (83.3%) | 55 (100%) | 0.641 |

SD: standard deviation.

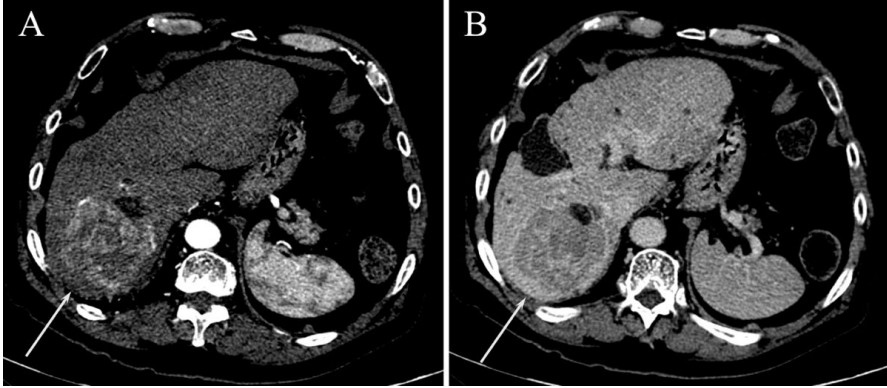

**Figure 2.** Axial CT images of a voluminous HCC nodule (white arrow) in segment 7 of the liver in a patient with NAFLD showing hyperenhancement in the arterial phase (**A**) and washout in the portal venous phase (**B**).

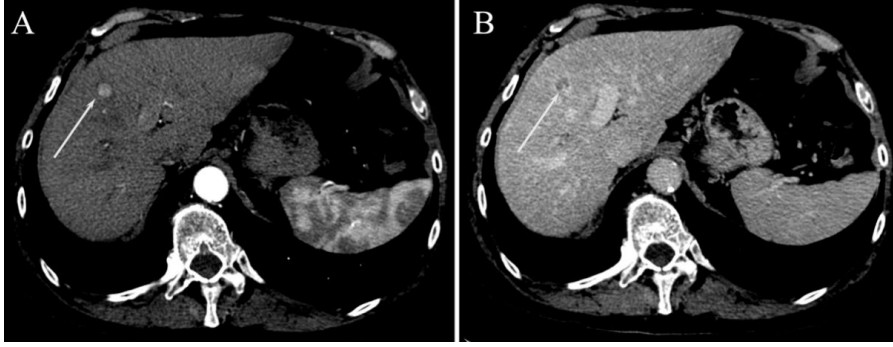

**Figure 3.** Axial CT images of a small HCC nodule (white arrow) in segment 8 of the liver in a patient with HCV showing hyperenhancement in the arterial phase (**A**) and washout in the portal venous phase (**B**).

Although a significant difference in tumor diameter was observed between the two groups, the size of the nodule did not correlate with the time to recurrence (HCV 9.1 months vs. NAFLD 8.2 months, $p = 0.778$), thus not impacting clinical outcomes.

*Surrogated Radiological Signs of MVI in NAFLD–HCC Patients and in HCV–HCC Patients*

In the entire study population (91 patients), the majority of patients underwent imaging using CT ($n = 77$), while MRI was used in the remaining 14 patients who had contraindications to CT (i.e., allergy to iodinated contrast medium and/or chronic kidney disease). Radiological signs of MVI were present in approximately half of the patients in each group. Both in NAFLD–HCC and the HCV–HCC patients, the most frequent suspicious sign of MVI was the presence of irregular tumor margins, followed by the presence of TTPVI (Figure 4). The presence of peritumoral enhancement (Figure 5) was an infrequent sign in both groups.

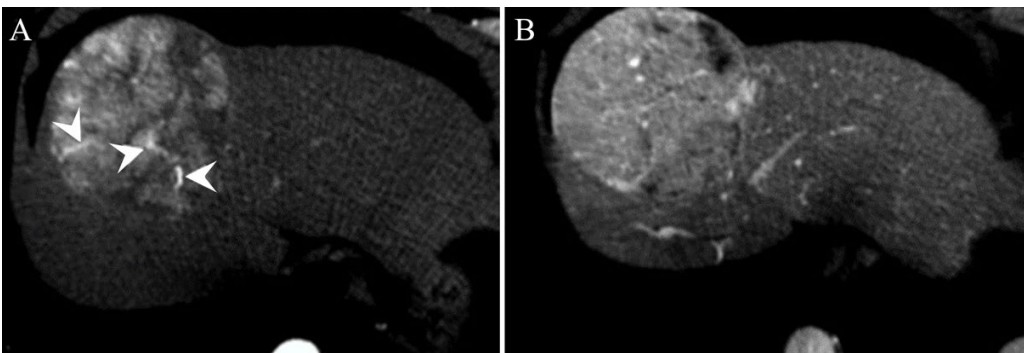

**Figure 4.** Axial CT images of a large HCC nodule in segment 4 of the liver in a patient with NAFLD showing the presence of TTPVI (white arrowheads) in the arterial phase (**A**) and the portal venous phase of the same patient (**B**).

The comparison between the groups showed statistically significant differences in nodule diameter, which was larger in NAFLD–HCC patients than in HCV–HCC patients (51 mm vs. 36 mm, respectively; $p = 0.004$) and a trend towards a higher prevalence of TTPVI in the NAFLD–HCC group as compared to the HCC–HCV group (38.9 vs. 20.0%, respectively; $p = 0.058$). On the other hand, there were no significant differences in terms of tumor margin irregularities or peritumoral enhancement (Table 2).

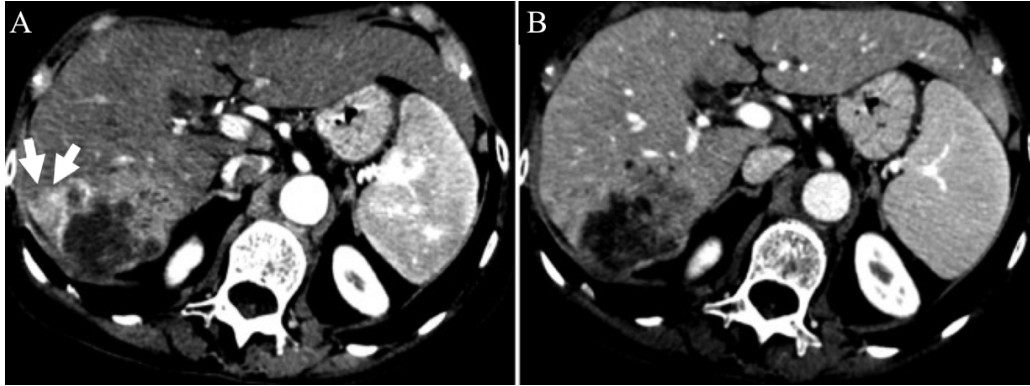

**Figure 5.** Axial CT images of an HCC nodule in segment 8 of the liver in a patient with NAFLD showing the presence of peritumoral enhancement in the arterial phase (white arrows) (**A**) and the portal venous phase of the same patient (**B**).

**Table 2.** Surrogated radiological signs of MVI in study population.

| MVI Parameter | NAFLD (*n* = 36) | HCV (*n* = 55) | *p* |
|---|---|---|---|
| Diameter (mm): mean ± SD | 51 (±23) | 26 (±16) | 0.004 |
| Irregular margins | 21 (58.3%) | 26 (47.3%) | 0.391 |
| TTPVI | 14 (38.9%) | 11 (20%) | 0.058 |
| Peritumoral enhancement | 4 (11.1%) | 6 (10.9%) | 1.000 |

SD: standard deviation.

Considering the differences in nodule diameter and, in particular, analyzing whether the TTPVI was dissimilar in nodules >50 mm or ≤50 mm, TTPVI was significantly greater in tumors >50 mm as compared to those that were less bulky (83.3% vs. 19%, respectively; $p < 0.0001$). A binary logistic analysis was carried out to clarify whether the higher prevalence of TTPVI in the NAFLD–HCC group depended on intrinsic tumor characteristics or on the greater nodule diameter (Table 3). With multivariate analysis, nodule diameter >50 mm was considered to be the only independent prognostic factor of TTPVI (HR: 21.3, 95% CI: 4.2–107.7, $p < 0.001$) (Table 4).

**Table 3.** Binary logistic regression analysis of factors associated with TTPVI.

| | Univariate Analysis | | Multivariate Analysis | |
|---|---|---|---|---|
| Variable | B (95% CI) | *p* | B (95% CI) | *p* |
| Diameter > 50 mm | 21.300 (4.200–107.700) | <0.001 | 21.300 (4.200–107.700) | <0.001 |
| Multinodularity | 1.365 (0.492–3.791) | 0.671 | | |
| Irregular margins | 1.428 (0.758–2.690) | 0.447 | | |
| Peritumoral enhancement | 1.119 (0.939–2.784) | 0.340 | | |

CI: confidence intervals.

In particular, the presence of TTPVI correlated with a higher, but not statistically significant, recurrence rate (88.0 vs. 73.8%, respectively; $p = 0.171$). However, there was a correlation between the presence of TTPVI and early recurrence (median TTR 11.4 months in patients without TTPVI vs. 4.1 months in patients with TTPVI, $p = 0.036$; Figure 6). In the multivariate Cox regression analysis, only the presence of TTPVI was confirmed to be an independent prognostic factor of recurrence, doubling the risk of recurrence (HR: 2.349, 95% CI: 1.369–4.032, $p = 0.002$). Considering the correlation between tumor diameter and TTPVI, whether or not the presence of nodules >50 mm was a risk factor itself for recurrence was assessed. However, in the present series, no differences were found between tumor size (> or ≤50 mm) and the TTR.

**Table 4.** Factors associated with time to recurrence.

| | Univariate Analysis | | Multivariate Analysis | |
|---|---|---|---|---|
| Variable | HR (95% CI) | *p* | HR (95% CI) | *p* |
| Age | 1.010 (0.989–1.018) | 0.399 | | |
| Male sex | 1.005 (0.993–1.759) | 0.121 | | |
| Size of the largest nodule | 1.020 (1.020–1.080) | 0.021 | 1.117 (0.787–1.585) | 0.352 |
| Multinodularity | 1.287 (0.959–1.727) | 1.333 | | |
| Irregular margins | 1.000 (1.000–1.000) | 0.318 | | |
| TTPVI | 2.058 (1.103–3.124) | 0.015 | 2.349 (1.369–4.032) | 0.002 |
| Peritumoral enhancement | 1.152 (0.865–1.534) | 0.334 | | |

HR: hazard ratio; CI: confidence intervals.

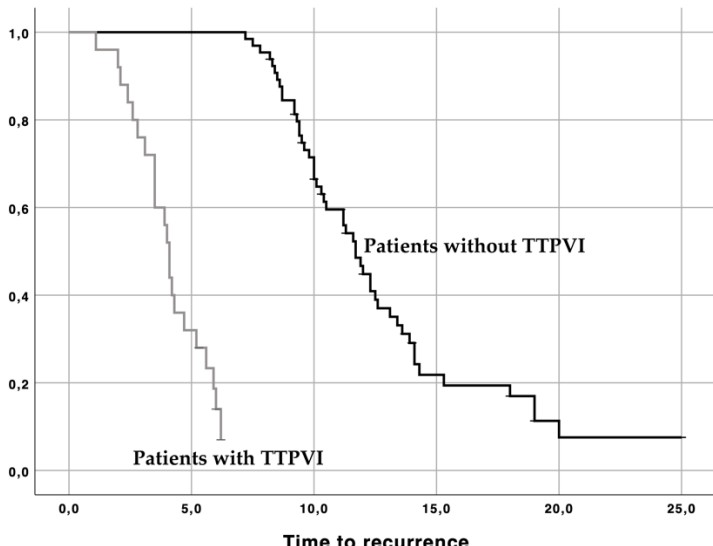

**Figure 6.** Kaplan–Mayer curves showing the recurrence rate for patients with and without TTPVI.

No correlations were found between TTR, and irregular tumor margins or peritumoral enhancement.

## 4. Discussion

The increasing incidence of metabolic syndrome is reflected in the changes in HCC epidemiology and etiology; in fact, to date, NAFLD represents the fastest growing cause of HCC in Western countries. In recent years, the natural history of HCC caused by NAFLD, as compared to other etiologies, has been investigated in depth. This study partially confirmed the evidence reported. In fact, in the present series, a slightly higher prevalence of NAFLD–HCC in males was demonstrated due to their predisposition to metabolic syndrome [5]. Moreover, at the time of diagnosis, the HCC nodules appeared larger in comparison to tumors caused by HCV liver disease, which can be explained by the lack of screening–surveillance tests in patients with NAFLD [5]. The absence of a surveillance program in these patients postpones the diagnosis, which frequently occurs at a more advanced stage, thus reducing the opportunity for curative treatment [32]. Despite this, the creation of scheduled surveillance using ultrasound or more accurate imaging techniques, such as abbreviated MRI protocols, is still a matter of debate. The more aggressive behavior of NAFLD–HCC has also been confirmed by the greater prevalence of radiological signs of MVI at diagnosis, some of which have a related prognostic factor. Regarding the clinical features, a greater prevalence of male gender and a larger size of the larger nodule in patients with NAFLD have been found.

In particular, in the present series, a correlation trend between the presence of TTPVI alone and the disease recurrence rate was found, although it was not statistically significant. The non-significance could probably be attributed to the short observation period. According to the results, the radiological signs of MVI correlated with a shorter time to recurrence after curative treatment. This aspect has recently been verified in an Asian study [33]. The advantage of the present series was that it proposed a differentiation between two different etiologies—HCV and NAFLD—showing that the radiological sign that most correlated with prognosis, specifically with the shorter time of disease recurrence, was TTPVI. Furthermore, although TTPVI was more frequent in larger nodules, in the present study, the presence of TTPVI was only related to the time to recurrence and not to the size of the nodule. One of the potential advantages deriving from the finding of radiological MVI was having prognostic information before treatment and, consequently, being able to guide the therapeutic choice towards more aggressive strategies that guaranteed better local control of the disease, compatibly with the degree of functionality of the residual liver [34]. In these cases, the advantage of using the curative techniques recently shown in the literature

may become even more evident [32]. Moreover, if the present results are confirmed, the pretreatment use of MVI would allow patients to be referred for better treatment, including complex treatments, such as chemo-embolization (which is also affected by stochastic and non-stochastic effects due to radiation exposure), and guiding the re-treatment strategy that today is not effectively guided by a laboratory score [35,36]. The presence of MVI could provide more information on the biological behavior of small HCC nodules that often are offered non-surgical curative treatment such as ablation. For instance, TTPVI in a small HCC nodule selected for ablation could require a more aggressive (surgical) approach, whereas a small HCC nodule selected for resection without TTPVI could benefit from percutaneous ablation. Further studies are needed to elucidate these hypotheses. The possibility of utilizing MVI in the pretreatment phase is perfectly in line with the new frontiers of modern hepatology, namely, tailored medicine, as for each treatment it will be possible to select the best candidate (MVI negative) or to stop futile re-treatments [37].

Finally, in the future, the use of radiomic features could improve the detection of MVI. In this regard, Li Yang et al. developed and validated a radiomic nomogram for the preoperative prediction of MVI in HCC [38]. In the present series, it was not possible to apply this nomogram because it was principally based on MRI findings, unlike the present experience in which the most utilized technique was CT.

The present study had several limitations. First, it was a retrospective, single-center study with a relatively small sample size due to the strict inclusion criteria. Second, the follow-up period was short. Future investigations should include a larger number of participants involving different centers, and a longer observational period, to confirm the findings of this study. Furthermore, all the images were evaluated by a single expert radiologist alone; however, the reproducibility of the imaging features of MVI was not an aim of the study, since it had already been established in previous studies [39]. Finally, considerable interobserver variability exists in the imaging assessment of MVI in HCC, even for more experienced radiologists [40].

In conclusion, in the present series, the NAFLD–HCC patients did not only have a larger tumor at diagnosis but also the more frequent presence of radiological signs of MVI as compared to the HCV–HCC patients. Unlike the size of the nodule alone, this presence is related to a more rapid recurrence after curative treatment. Taken together, these data recognized the prognostic value of radiological MVI signs for the first time in Western cases and suggested seeking better and possible treatment for achieving the best local disease control. Furthermore, this triple association between TTPVI, MVI, and the genetic profile could allow the use of imaging to reconstruct HCC gene expression programs in the future, thus creating a non-invasive molecular portrait of the tumor with the aim of establishing targeted therapies in the era of tailored medicine.

**Author Contributions:** Conceptualization, M.R., A.P., A.G. and R.G.; methodology, M.R., A.P. and A.G.; validation, M.R. and R.G.; formal analysis, M.R., A.P. and A.G.; investigation, A.P., N.B., F.A. and B.S.; resources, M.R., A.G. and R.G.; data curation, A.P. and G.M.; writing—original draft preparation, M.R., A.P. and F.A.; writing—review and editing, M.R. and N.B.; visualization, M.R., F.T. and A.G.; supervision, M.R., A.G. and R.G.; project administration, M.R. and F.T.; funding acquisition, M.R., A.G. and R.G. All authors have read and agreed to the published version of the manuscript.

**Funding:** This research received no external funding.

**Institutional Review Board Statement:** The study was conducted in accordance with the Declaration of Helsinki and approved by the Institutional Review Board of IRCCS Azienda Ospedaliero-Universitaria di Bologna.

**Informed Consent Statement:** Informed consent was obtained from all subjects involved in the study.

**Data Availability Statement:** The data presented in this study are available upon request from the corresponding authors.

**Conflicts of Interest:** The authors declare no conflict of interest.

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
