# Peer review of "Radiological Features of Microvascular Invasion of Hepatocellular Carcinoma in Patients with Non-Alcoholic Fatty Liver Disease"

_gastroent, doi:10.3390/gastroent13030028_

Round 1

Reviewer 1 Report

The revised manuscript is now improved, and the Authors have satisfactorily addressed all the raised points. The manuscript can now be accepted.

Reviewer 2 Report

Dear Authors, 

Thank you for considering all the comments. I recognized the efforts in improving the manuscript. I have no additional comments.

This manuscript is a resubmission of an earlier submission. The following is a list of the peer review reports and author responses from that submission.

Round 1

Reviewer 1 Report

In this interesting study, the authors aimed to evaluate the presence and the prognostic value of the radiological signs of microvascular invasion (MVI) of hepatocellular carcinoma (HCC) in patients with non-alcoholic fatty liver disease (NAFLD).

Patients (91 patients) with de novo HCC or HCC recurrence occurring at least 2 years after the last treatment in NAFLD (36 patients) or for hepatitis C virus (HCV) liver disease (55 patients) were studied after surgical HCC treatments (liver resection and transplantation) to obtain the anatomopathological confirmation of MVI.

They found that NAFLD-HCC nodules were larger as compared to HCV-HCC (51 mm vs. 36 mm, p=0.004) and showed a higher prevalence of TTPVI (38.9 vs. 20.0%, p = 0.058). At multivariate analysis, nodule size >50 mm was the only independent prognostic factor of TTPVI (Hazard Ratio: 21.3, 95% Confidence Interval: 4.2-107.7, p<0.001), and the presence of TTPVI were confirmed to be the only independent prognostic factors of recurrence (Hazard Ratio: 2.349, 95% Confidence Interval: 1.369-4.032, p=0.002). No correlations were found between TTR, and irregular tumor margins or peritumoral enhancement.

They concluded that NAFLD-HCC patients had larger tumors at diagnosis and showed a more frequent presence of radiological signs of MVI as compared to the HCV-HCC patients. The MVI was related to a more rapid recurrence after curative treatments.

The study provides clinically significant findings, however, some points should be addressed.

-All the patients included in the present study received anatomopathological 153 confirmation of MVI: please, provide which histological criteria were used to assess MVI.

-English would require revisions ("dimensions" should be replaced with nodule size or nodule diameter).

Reviewer 2 Report

-- I think it was a great paper. I'd be better to have a multicenter study or larger N but it is an interesting read. 

Reviewer 3 Report

General comments

This study compared the presence of some radiological features of microvascular invasions in patients with NAFLD and HCV. The study evaluates an interesting and relevant topic. However, this study is limited by the assessment both CT (77 patients) and MRI (14 patients), lack of inter-reader assessment for qualitative imaging features, and lack of comparison with the reference standard (histopathological diagnosis of MVI since all patients were surgically confirmed).

Specific comments

Abstract:

-Please specify if the patients were imaged with contrast-enhance CT and/or MRI.

Introduction:

-Page 2, lines 46-58: The description of NAFLD is quite long and may be shorten to be more focus on the diagnosis of HCC and the clinical relevance of microvascular invasion detection for the patients’ management.

-Page 2, line 88: Please define TTPVI when used for the first time in the abstract.

-Page 2: It should be noted that the definition of imaging features associated with MVI has been very heterogeneous in the radiology literature and some studies reported that those features have poor inter-observer reproducibility (Radiology. 2020 Dec;297(3):573-581).

Materials and Methods:

-Page 3, line 99: “after the last treatment for NAFLD (cases)”. Which type of treatment were considered for NAFLD? Please clarify.

-Page 3, lines 98-104: Was any exclusion criteria considered for this study? Why both CT and MRI were considered? A flowchart of the patients population could have been provided.

-Page 3, lines 111-113: The study protocol must be more detailed for this type of study. Type of scanners, imaging protocol and type of contrast agent used for imaging acquisition must be at least mentioned in the methods. The reference #30 “Congenital diseases of the thoracic aorta. Role of MRI and MRA” looks not appropriate for a liver imaging protocol reference in this type of study. Please revise.

-Page 3, lines 113-115: A single radiologist evaluated all the images. A second radiologists could have been involved to assess the inter-reader agreement considering the subjective interpretation of some imaging features.

-Consider to provide a dedicated paragraph on the histopathological reference standard for this study.

Results:

-Page 4, lines 149-152: Please specify the number of patients with cirrhosis and the staging of chronic liver disease (Child-Pugh class).

-Page 4, lines 153-157: It is unclear the final number of patients diagnosed with MVI at histopathological analysis. Please specify.

-Page 5, lines 174-186: In this study the imaging signs of MVI are compared between NAFLD and HCV. However, there is not direct comparison with the reference standard of histopathology. Therefore, it is unclear if those signs can effectively used to predict MVI in NAFLD.

-Page 5, lines 176-177: “MRI was used in the remaining 14 patients who had contraindications to CT”. Please define the contraindications for CT. Beside pregnancy and some major allergic reaction to contrast agents, there are no major contraindications to CT.

-Page 5: Could different imaging modality affect the prevalence of radiological signs of MVI? It would have been more appropriate to select only contrast-enhanced CT to limit the difference between the imaging modalities that could affect the analysis.

Discussion:

-Page 7, lines 264-266: When referring to the tailored medicine for HCC, do the Authors mean systemic treatment in patients with more aggressive HCCs?

-Page 7, lines 273-277: Consider to discuss the limitations regarding the different imaging modalities and the lack of inter-reader analysis.

Tables:

-Table 1: Please provide the number of patients with cirrhosis the Child-Pugh class. Please include also the histopathological analysis (MVI) in the table.

Figures:

-Please provide at least one case showing the presence of radiological sign of MVI (especially TTPVI and peritumoral enhancement) on CT and MRI.

-Consider to provide Kaplan-Mayer curves for patients with and without TTPVI.